

# High anteromedial insertion reduced anteroposterior and rotational knee laxity on mid-term follow-up after anatomic anterior cruciate ligament reconstruction

Xiaohan Zhang[1,*], Yi Qian[2,*], Feng Gao[2], Chen He[2], Sen Guo[2], Qi Gao[1] and Jingbin Zhou[2]

[1] School of Sport Medicine and Rehabilitation, Beijing Sport University, Beijing, China
[2] National Institute of Sports Medicine, Beijing, China
[*] These authors contributed equally to this work.

Corresponding authors
Qi Gao, fevok@sina.com
Jingbin Zhou, jingbinzhou@163.com

## ABSTRACT

**Background**. The position of the femoral insertion has a great influence on the laxity of the knee joint after ACLR, especially for rotational laxity.

**Purpose**. To compare the effects of different femoral tunnel positions on knee stability after arthroscopic anterior cruciate ligament reconstruction (ACLR).

**Methods**. The clinical outcomes of 165 patients after autograft ACLR were analyzed retrospectively. The patients were separated into three groups according to the position of the femoral tunnel, as follows: low center (LC) group, 53 patients; high center (HC) group, 45 patients; and high anteromedial (HAM) group, 67 patients. The side-to-side differences (SSDs) in anteroposterior knee laxity measured using a KT-2000 arthrometer and the pivot shift test (PST) pre- and postoperatively were compared among the three groups and analyzed.

**Results**. After 5 years postoperatively, the SSD in the anteroposterior knee laxity in the three groups was significantly decreased postoperatively compared with preoperatively in knees; meanwhile, the negative PST rate was significantly increased in the three groups. The postoperative SSD in anteroposterior knee laxity was significantly increased in the HC group compared with the LC and HAM groups ($1.5 \pm 1.3$ VS $1.0 \pm 1.1$ VS $1.0 \pm 1.0$, $P < 0.05$). The negative postoperative PST rate was higher in both the LC and HAM groups than in the HC group (84.9% VS 91.0% VS 71.1%, $P < 0.05$), and there was no significant difference in the negative PST rate between the LC and HAM groups (84.9% VS 91.0%, $P > 0.05$). The negative postoperative PST rate was significantly higher in the HAM group than in the LC and HC groups for patients with a high degree of laxity preoperatively (31.3% VS 3.3% VS 14.4%, $P > 0.05$).

**Conclusion**. Patients in HAM group showed better control over anteroposterior laxity, rotational laxity, and subjective knee function compared to other groups post operation. Therefore, the HAM point is the closest to the I.D.E.A.L point concept, and is recommended as the preferred location for the femoral tunnel in ACLR.

## INTRODUCTION

Anatomic anterior cruciate ligament reconstruction (ACLR) allows better recovery of knee stability due to the ability of the graft to be placed closer to its original insertion (*Sasaki et al., 2021*). Knee laxity often occurs after ACLR (*Wyatt et al., 2014*), and the main factors include the graft source and graft fixation method (*Hsu & Wang, 2013*), the position of the bone tunnel (*Hensler et al., 2013*), whether the meniscus is injured (*De Phillipo et al., 2020*) and whether postoperative adhesions develop (*Cristiani et al., 2019*). The position of the bone tunnel is a factor that surgeons can adjust to avoid knee laxity after ACLR. The position of the femoral insertion has a great influence on the laxity of the knee joint after ACLR, especially for rotational laxity (*Bernard et al., 2020*). The attachment area of the femoral insertion of the ACL is large and eccentric, and its center can be located in different positions, resulting in different biomechanical insertion points (*Fujimaki et al., 2016*). *Van der List et al. (2017)* divided the ligament fibres at different insertion points of the ACL into the anteromedial (AM) bundle and the posterolateral (PL) bundle, or the bundle between AM and PL called the central (C) bundle. According to the theory of direct and indirect insertion points, the position of each bundle can be divided into superior and inferior areas. Meanwhile, biomechanical testing has shown that a femoral insertion point at the high AM (HAM) bundle is the most isometric and bears a greater biomechanical load (*Sasaki et al., 2021*). Therefore, since the establishment of the anatomic theory for ACLR, in clinical practice, the position of the femoral insertion has changed from the anatomic center position to become more eccentric (*Lee et al., 2015*). However, determining which position yields better outcomes on clinical follow-up requires further investigation.

The purpose of this study was to retrospectively analyze the influence of the three positions on knee laxity after ACLR. The hypothesis of this study was that a HAM femoral tunnel may reduce anteroposterior and rotational knee laxity after ACLR.

## MATERIAL AND METHODS

### Participants

The study was approved by the Ethics Committee of the National Institute of Sports Medicine (No.: 202219). Given the retrospective design of this study, the need to obtain written informed consent from the included patients was waived. From 2012 to 2017, a total of 329 ACL-injured patients underwent ACLR at the Sports Trauma Surgery Department of the Sports Hospital of the General Administration of the National Institute of Sports Medicine. Using the following inclusion criteria, a total of 165 patients were included in the study, consisting of 104 males and 61 females with an average age of $33.2 \pm 10.5$ years. The inclusion criteria were as follows: complete ACL rupture confirmed by arthroscopic exploration; no ACL rerupture; cooperate with this study. The exclusion criteria were as follows: history of knee surgery; other concomitant ligament injuries, such as posterior cruciate ligament injury, lateral collateral ligament injury or medial collateral ligament injury (over 3 degrees); abnormal lower limb alignment (such as abnormal Q angle); tibial

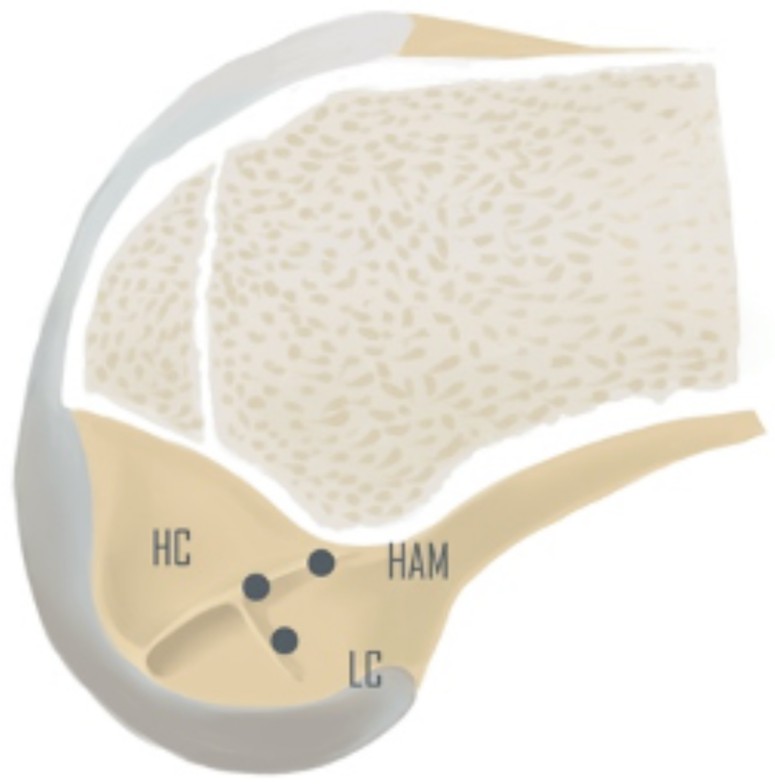

**Figure 1** **The location of the femoral tunnel.** LC (low center insertion), HC (high center insertion), HAM (high anteromedial insertion).

posterior tilt angle greater than 17 degrees; and generalized ligamentous laxity. The patients underwent follow-up for 5 years after the operation.

According to the convention for femoral tunnel placement adopted in different eras, we divided patients from different periods into the following three groups: the low center (LC) group; the high center (HC) group; and the high anteromedial (HAM) group (Fig. 1). The distribution of the patients was as follows: LC group, 53 cases; HC group, 45 cases; and HAM group, 67 cases. All patients received surgical treatment within one week after diagnosis. All patients underwent anteroposterior and rotational knee laxity tests before the operation. The anteroposterior laxity of the knee joint was tested using a KT-2000 knee arthrometer, and the index was the bilateral difference at 134 N. Rotational laxity was tested by the pivot shift test (PST). All patients were followed up using telephone invitations and outpatient testing methods, and the test data were obtained from the last follow-up.

## Surgical technique

All operations were performed by the same senior surgeon. The patient was placed in a supine position, the feet were relatively fixed, the knees were flexed 90 degrees, and the tourniquet was set to 300 mmHg. During arthroscopic exploration, many kinds of identified meniscal injuries required repair. If a bucket-handle injury of the meniscus was
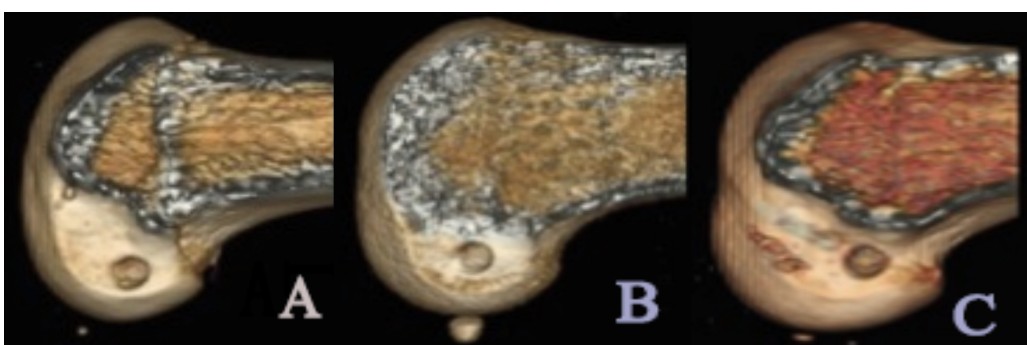

**Figure 2** **Postoperatively 3D CT reconstruction to verify the positioning of the femoral tunnel.** (A) Low center (LC) insertion; (B) high center (HC) insertion; (C) high anteromedial (HAM) insertion.

observed, the vertical injury was sutured in an inside-out manner, the ramp area of the medial meniscus was repaired with an all-inside method, and the posterior root of the lateral meniscus was repaired with an all-inside method or reconstructed using a single bone tunnel. The hamstring was harvested and made into a 4- to 8-fold autograft that was no less than eight mm in size. The knee was flexed to 150 degrees, and osseous landmarks of the lateral wall of the intercondylar fossa were observed through the medial-assisted arthroscopic approach and used for localization, meanwhile, the positioning was confirmed by intraoperative X-ray fluoroscopy. The femoral insertion point in the LC group was placed at the anatomic center point, which is inferior to the resident's ridge and slightly posterior to the bifurcate ridge (we used this femoral insertion point from Feb. 2010 to Apr. 2014). The femoral insertion point in the HC group was placed at the superior edge of the insertion site of the anatomic center, which is on the resident's ridge and superior to the LC (we used this femoral insertion point from May 2014 to Mar. 2016). In the HAM group, the center of the AM bundle and the superior edge of the insertion point were placed on the resident's ridge, 5-6 mm from the posterior edge of the femoral condyle cartilage (we used this femoral insertion point from Jun. 2016 to Dec. 2017). The tibial insertion point was placed at the center of the tibial insertion point stump. The femoral end was fixed with a fixed-loop cortical suspension device, and the tibial side was fixed with a sheathed extrusion nail. Immediately after surgery, three-dimensional CT reconstruction was used to verify the positioning of the femoral tunnel (Fig. 2).

## Postoperative rehabilitation and return to sports

A protective hinged brace was used for 6 weeks after the operation. All patients were followed according to the standardized rehabilitation protocol after ACLR in our department. The early range of motion was limited, with the passive range of motion being limited to 90 degrees for 2 weeks after the operation. The patients performed light weight-bearing activity with the assistance of crutches 4 weeks after the operation. Six weeks after the operation, the brace was replaced with a shorter soft brace, and patients were advised to engage in low to moderate intensity aerobic exercise at 6 months after the operation, such as jogging or swimming, and return to contact sports at 10–12 months after the operation.

**Table 1  Comparison of preoperative general data among the three groups (age unit: years, KT2000 unit: mm).**

|  | LC | HC | HAM | *P* Value |
|---|---|---|---|---|
| Age | $33.2 \pm 8.3$ | $35.0 \pm 11.0$ | $31.5 \pm 11.5$ | >0.05 |
| M: F | 34:19 | 27:18 | 43:24 | |
| KT2000 SSDs | $4.8 \pm 1.0$ | $4.8 \pm 0.7$ | $4.6 \pm 1.6$ | >0.05 |
| PST P ratio | 100% | 100% | 98.5% | >0.05 |
| Graft failure | 4(7.5%) | 10(22.2%) | 3(4.5%) | >0.05 |
| Meniscal injury | 26(49.1%) | 25(55.6%) | 44(65.7%) | >0.05 |
| Follow-up time | $5.0 \pm 0.1$ | $5.0 \pm 0.1$ | $5.0 \pm 0.2$ | >0.05 |

## Laxity evaluation

The same surgeon has done all examinations. The anteroposterior laxity of the knee joint was measured by the KT-2000 arthrometer, with the side-to-side difference (SSD, mm) when the knee was at 30 degrees of flexion and subjected to 134 N as the index. Rotational knee laxity was measured under non-anesthesia (*Nakamura et al., 2017*) by the PST and graded using 3 degrees, as follows: first degree, obvious sliding of the femoral tibial joint; second degree, obvious dislocation of the femoral tibial joint; and third degree, obvious interlocking of the femoral tibial joint.

## Statistical analysis

Data were analyzed by SPSS 25.0. The Shapiro–Wilk test showed a *P* value of 0.078, indicating a normal distribution. T tests were used to compare the anteroposterior laxity of the knees in the three groups between before and after the operation. The chi-square test was used to compare categorical variables, including the positive and negative PST rates, among the three groups before and after the operation. One-way ANOVA was used to compare the rates of lateral and medial meniscal injury and repair surgery among the three groups. The rank-sum test was used to compare the distribution of the preoperative and postoperative PST results in each group. Data are expressed as the mean $\pm$ standard deviation, and $P < 0.05$ was considered to indicate a statistically significant difference.

## RESULTS

The preoperative KT-2000 test results of the three groups were similar, and the proportions of positive PST results were similar (Table 1). Other demographic data also showed no statistical difference.

The KT-2000 test after ACLR showed that the anteroposterior laxity in the three groups was significantly improved compared with that before surgery ($P < 0.05$), but the effect in the HC group was worse than that in the LC and HAM groups ($P < 0.05$). The postoperative anteroposterior laxity between the LC and HAM groups were similar (Tables 2 and 3).

The preoperative distribution of PST results among the three groups was similar, and all groups showed significant improvement after the operation ($P < 0.05$) (Table 4). The positive PST rate of the HC group was significantly higher than the HAM group ($P < 0.05$), meanwhile, the HC group is similar to the LC group ($p > 0.05$). However, the postoperative

**Table 2 Comparison of the difference in the anteroposterior laxity of the knees between the three groups (unit: mm). Pre-operative (Pre-op), Post-operative (Post-op).**

|  | Pre-op | Post-op | P Value |
|---|---|---|---|
| LC | $4.8 \pm 1.0$ | $1.0 \pm 1.1$ | $<0.05$ |
| HC | $4.8 \pm 0.7$ | $1.5 \pm 1.3$ | $<0.05$ |
| HAM | $4.6 \pm 1.6$ | $1.0 \pm 1.0$ | $<0.05$ |
| P Value | $>0.05$ | $<0.05$ |  |

**Table 3 Comparison of the change of the number of cases between the three groups.**

| KT2000 SSDs | <3 mm | | 3–5 mm | | >5 mm | |
|---|---|---|---|---|---|---|
|  | Pre-op | Post-op | Pre-op | Post-op | Pre-op | Post-op |
| LC | 0 | 48* | 42 | 5[*] | 11 | 0[*] |
| HC | 0 | 36* | 40 | 9[*] | 5 | 0[*] |
| HAM | 6 | 63* | 47 | 4[*] | 14 | 0[*] |

Notes.
An asterisk (*) indicates significant difference between pre-operation and post-operation, $P < 0.05$.

**Table 4 Comparison of 165 cases (%) in three groups of PST.**

|  | LC | | HC | | HAM | |
|---|---|---|---|---|---|---|
|  | Pre-op | Post-op | Pre-op | Post-op | Pre-op | Post-op |
| N | 0 (0%) | 45 (84.9%) | 0 (0%) | 32 (71.1%) | 1 (1.5%) | 61 (91.0%) |
| Grade I | 46 (86.7%) | 8 (15.1%) | 34 (75.6%) | 10 (22.2%) | 45 (67.2%) | 6 (9.0%) |
| Grade II | 7 (13.3%) | 0 (0%) | 11 (24.4%) | 3 (6.7%) | 21 (31.3%) | 0 (0%) |
| Grade III | 0 | 0 | 0 | 0 | 0 | 0 |

positive PST rate between the HAM and LC groups postoperatively was similar ($p > 0.05$).

Postoperative PST results were compared among patients with a high degree of instability (degree II–III) preoperatively, and a significant difference in postoperative PST results was found in patients with the three groups ($P < 0.05$). Further pairwise comparison revealed that the postoperative PST result of patients with femoral tunnels in the HAM point was significantly better than those of patients with femoral tunnels in the HC and LC point ($P < 0.05$), with no significant difference between the HC and LC group ($p > 0.05$) (Table 5). There was one patient who was particularly unique, with a preoperative positive PST of 1 degree, but graft failure with a postoperative positive PST of 2 degree.

According to the diagnostic criteria (*Roethke et al., 2023*) for ACL graft failure, which includes a KT-2000 SSD $\geq 3$ mm and/or a positive pivot shift test, among the 165 patients included in the study, 17 cases experienced graft failure within 5 years after surgery, resulting in a failure rate of 10.3%. Of these cases, 4 failures (7.5%) were observed in the LC group, 10 failures (22.2%) in the HC group, and 3 failures (4.5%) in the HAM group. There were no significant differences among the three groups in the preoperative incidence

**Table 5 Postoperative PST results for patients with high preoperative PST.**

| PST | LC (cases) | HC (cases) | HAM (cases) |
|---|---|---|---|
| N | 2 | 1 | 16 |
| Grade I | 5 | 8 | 5 |
| Grade II | 0 | 2 | 0 |
| Grade III | 0 | 0 | 0 |

**Table 6 Comparison of 165 cases (%) in three groups of lateral and medial meniscus injuries and repair.**

| | LC | | HC | | HAM | |
|---|---|---|---|---|---|---|
| | Pre-op | Post-op | Pre-op | Post-op | Pre-op | Post-op |
| MM | 12 (22.6%) | 0 (0%) | 15 (33.3%) | 0 (0%) | 25 (37.3%) | 0 (0%) |
| LM | 14 (26.4%) | 0 (0%) | 10 (22.2%) | 0 (0%) | 19 (28.3%) | 0 (0%) |
| Operation | 26(49.1%) | | 25(55.5%) | | 44(65.7%) | |
| Cure | 26(100%) | | 25(100%) | | 44(100%) | |

of lateral and medial meniscal injury and repair ($p > 0.05$). After five years, the meniscus healed well in all patients who underwent meniscal repair surgery (Table 6).

## DISCUSSION

According to five years of clinical follow-up results, compared with the HC group, the HAM and LC groups showed better improvement in the anteroposterior and rotational laxity of the knee joint, especially for patients with high degree pivot shift. Choosing the HAM point can reduce the postoperative laxity.

It is well known that anatomic ACLR allows the graft to bear greater internal stress and *in situ* force and thus better limits anteroposterior laxity (*Cain Jr et al., 2017*). This study confirmed that choosing the reconstruction of the femoral tunnel position in the anatomical region of the femoral insertion can significantly improve the control of anteroposterior laxity of the knee joint. As anatomic ACLR technology has progressed, the position of the femoral insertion has transitioned from a low-center point to a high-center point and finally to a high anteromedial point (*Lee et al., 2015*). In all three groups of patients, the postoperative anteroposterior laxity SSD was $1.0 \pm 1.1$ mm for the LC group, $1.5 \pm 1.3$ mm for the HC group, and $1.0 \pm 1.0$ mm for the HAM group, all less than three mm. *Musahl et al. (2022)* pointed out that the laxity of the knee joint is related to the inclination angle of the graft, and a lower plateau of the graft can better control anteroposterior and rotational laxity. According to the findings of this study, the graft inclination angle at the HC point is relatively larger than that at the LC point and HAM point, and its control of anteroposterior laxity is relatively worse, which also proves the research of Musahl et al. However, SSD of anteroposterior laxity in the HC group was still less than three mm after five years, indicating that the use of the HC point as the femoral insertion is also appropriate in terms of controlling anteroposterior laxity.

Anatomical reconstruction can better restore the biomechanical function of the knee joint through the evaluation of rotational laxity by PST (*Yasuma et al., 2020*; *Komzak et al.,*

*2021*; *Zampeli et al., 2014*). This study found that all three groups significantly improved the rotational laxity of the knee joint after surgery compared with preoperative status, and the negative rates of PST after 5 years were 84.9%, 71.1%, and 91.0% in the LC, HC, and HAM groups, respectively. However, the result also shows the differences in the three femoral tunnels in controlling the rotational laxity of the knee joint. Compared with the LC group, although the positive rate of PST in the HAM group after surgery was not significantly different, its statistical figures showed that the positive rate was lower than that in the LC group. Compared with the HC group, the HAM group had a significant difference in the positive rate of PST after surgery, indicating that the HAM point can better control rotational laxity. The LC point is located at the center of the femoral insertion of the ACL (*Youm et al., 2014*), and the single-bundle reconstruction surgery is performed at this position (*Andrei, Niculescu & Popescu, 2016*), this method can better control knee joint rotation, thereby resisting greater internal stress (*Hoshino et al., 2018*), but this may cause changes in graft characteristics and even increase the failure rate of the graft (*Roach et al., 2023*). Therefore, *Takahashi et al. (2022)* proposed that the resident's ridge is the direct femoral endpoint of ACL and bears the main stress, which has better isometry in knee extension and flexion processes and reduces the failure rate of the graft (*Hammarstedt et al., 2023*). The HC point applies this concept and moves the LC point to the position of the resident's ridge (*Thanasrisuebwong et al., 2023*). According to the 5-year follow-up results, there was no significant difference in the positive rate of PST between the HC and LC groups, indicating that there was no significant advantage in controlling rotational laxity with the HC point compared with the LC point. However, contrary to the previous theory, the HC group had a far higher graft failure rate than the other two groups after 5 years, which may be the reason why the HC point was abandoned in a short time.

The HAM point is located in the posterior and upper 25% of a quadrant (*Kadija et al., 2017*), and *Bedi et al. (2010)* and *Hoshino & Fu (2010)* used X-ray fluorescence to determine the standard AM position, and the HAM point is placed on the upper edge of the AM bundle insertion, so the bone position is higher and forward than in the LC and HC groups. According to the biomechanical study by *Van der List et al. (2017)*, the posterior part of the fiber bundle (*i.e.,* the HAM bundle) shows the best isometric characteristics and bears greater biomechanical load. The results of this study were similar, indicating that after 5 years, the HAM group had a higher negative rate of PST than the other two groups, and in the absence of significant axial displacement, the HAM group performed better in controlling rotational laxity, which was confirmed by comparing preoperative and postoperative data. According to the theory of ACL anatomic reconstruction, the ideal femoral tunnel insertion for ACLR is eccentric, with the optimal force not located at its physical center. The current internationally recognized concept of the femoral tunnel for ACLR, as proposed by *Pearle, McAllister & Howell (2015)*, is the I.D.E.A.L (Isometric, in the Direct fibers, Equidistant and Eccentric, Anatomic, and Low in tension) point. This concept provides a biomechanically equidistant eccentricity, the original anatomical ACL position, and the lowest laxity and tension when the knee is extended. Based on the results of our research, we believe that the position of the HAM point is closer to the I.D.E.A.L point (*Inderhaug et al., 2017*; *Naghibi et al., 2020*). The graft in this position has a better

healing environment, easier achievement of optimal biomechanical function, lower graft failure rate and improved laxity at 5 years after ACLR.

Meniscal injury is an important factor affecting the anteroposterior and rotational laxity of the knee after ACLR (*Nakamura et al., 2021*; *Okazaki et al., 2020*). According to the statistical results of this study, there were no significant differences in the incidence, surgical treatment rate or cure rate of meniscal injury among the three groups. Therefore, the occurrence of meniscal injury had no influence on the results of this study.

This study had the following limitations: (1) This was a retrospective study, and the selection of cases may have been biased. (2) This study had a mid-term follow-up duration, and long-term follow-up is still needed. Future research needs to comprehensively evaluate these influencing factors. (3) Although there was no statistically significant difference in results and grading between non-anesthesia relocation tests and anesthesia relocation tests, their consistency and objectivity can still be improved. (4) Since our study was a mid-term follow-up, changes in physical functioning or psychological status (*Philippot et al., 2019*) of the included subjects may occur over the course of 5 years, which could potentially cause some bias in the results. (5) Lack of subjective knee evaluation.

## CONCLUSION

According to the mid-term clinical follow-up results of 5 years, ACLR using three different femoral tunnel positions can effectively control the anteroposterior and rotational laxity of the knee joint after ACL injury. Compared with the HC and LC groups, the HAM group performed better in controlling the anteroposterior and rotational laxity of the knee joint, and the patients had better graft failure rate after surgery. Especially for patients with high rotational laxity, the HAM position can reduce the postoperative rotational laxity. Based on the conclusion of this study, we believe that the HAM point is closest to the position recommended by the I.D.E.A.L concept, so we recommend HAM as the preferred femoral tunnel position for ACL reconstruction.

### Funding
This project was supported by the Hainan Province Clinical Medical Center. The funders had no role in study design, data collection and analysis, decision to publish, or preparation of the manuscript.

### Grant Disclosures
The following grant information was disclosed by the authors:
Hainan Province Clinical Medical Center.

### Competing Interests
The authors declare there are no competing interests.

### Author Contributions
- Xiaohan Zhang performed the experiments, analyzed the data, prepared figures and/or tables, and approved the final draft.

- Yi Qian performed the experiments, analyzed the data, authored or reviewed drafts of the article, and approved the final draft.
- Feng Gao performed the experiments, prepared figures and/or tables, and approved the final draft.
- Chen He performed the experiments, prepared figures and/or tables, and approved the final draft.
- Sen Guo performed the experiments, authored or reviewed drafts of the article, and approved the final draft.
- Qi Gao conceived and designed the experiments, authored or reviewed drafts of the article, and approved the final draft.
- Jingbin Zhou conceived and designed the experiments, prepared figures and/or tables, and approved the final draft.

## Human Ethics

The following information was supplied relating to ethical approvals (*i.e.*, approving body and any reference numbers):

The Ethics Committee of the National Institute of Sports Medicine granted Ethical approval to carry out the study within its facilities (No.: 202219)

## Data Availability

Raw data are available as a Supplemental File.

## Supplemental Information

Supplemental information for this article can be found online at http://dx.doi.org/10.7717/peerj.15898#supplemental-information.

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
