# Peer review of "High anteromedial insertion reduced anteroposterior and rotational knee laxity on mid-term follow-up after anatomic anterior cruciate ligament reconstruction"

_PeerJ, doi:10.7717/peerj.15898_

## Round 0.1 · original submission · Minor Revisions

Dear Dr. Zhou,

Thanks for the submission to PeerJ. We sent it out for peer review and most of the reviewers found that this 5 years follow-up retrospective research has been well designed and clearly written, the dataset and the conclusion of which is helpful to improve the clinical work of ACL reconstruction.

Nevertheless, there are still some minor concerns that need to be clarified. Especially, some of the descriptions may need to be elaborated. Therefore, I would like to ask you to address the reviewers' comments accordingly.

Please let me know if anything is not clear to you during the revision.

Best,
Jian Song

Reviewer 1 ·

Basic reporting

This paper describes the effect of three different femoral insertion of grafts during ACL reconstruction 5 years after surgery. There is no doubt of interest to orthopedic surgeons as optimal positioning of the graft is an important goal and ongoing challenge in ACL reconstruction. However, I am still confused by some of the descriptions in this article. If the author can clarify my doubts, I suggest that they revise and publish the paper.

Experimental design

no comment

Validity of the findings

no comment

Additional comments

1. L89: By multiple ligament laxity do you mean generalised joint hypermobility?
2. L90:Were the KT 2000 measures taken at the five-year follow-up? This is not clear. Were any other measures taken at the five-year follow -up?
3. L126: At what stage post-surgery was the 3D CT reconstruction done?
4. L258: What is mean of the extra-articular structure?

Reviewer 2 ·

Basic reporting

I thank the authors for their work and submission. The idea of this research is good, and the results could be important in the clinical practice. I think the main finding of this paper is reliable. However, there are still some mistakes in this paper. Despite these mistakes this paper may influence surgical approaches and I consider it of sufficient scientific merit to, once revised, be considered for publication in the Peer J.

Experimental design

No comment

Validity of the findings

No comment

Additional comments

1. L82: What was the size of the original 2012-2017 cohort? How many patients were excluded from the study?
2.L111: Do you use Accessory Medial Portal to explain the surgical technique? And how to fix the graft on the tibial part (Pre-tension, knee angle).
3.What is your advice in the future regarding the creation of femoral tunnel? You’d better put them in Conclusion.

·

Basic reporting

The idea of this article is clear, and the research design and methods are reasonable, so the research results appear to be reliable. Perhaps it may have a good effect on the improvement of ACL reconstruction results, so it has strong clinical guiding significance. Therefore, I believe that this paper can be considered for publication in Peerj.

Experimental design

1.The grouping of the insert for the three types of ACLR in this article was created based on mainstream surgical methods at different times, which means that surgeries in different research groups are not performed in parallel at the same time. However, as a retrospective study, this is also an inherent flaw. What I want to understand is how the author controls the quality of the research and avoids bias in results caused by the advancement of surgical techniques by surgeons at different times.
2. This study was followed up for 5 years, and I would like to know if each patient was followed up exactly 5 years after surgery? Or is the patient scheduled for follow-up during a certain period of time?

Validity of the findings

1. L80: Can cartilage lesion be included or excluded as a criterion.
2. L120: Can the author provide evaluation criteria and methods for CT evaluation of the insert position? In addition, in most cases, CT scanning is not necessary after ACLR surgery, so is the additional cost for patients voluntary?
3. L147: The lack of functional scoring is the weakness of this study. Functional scoring is crucial for us to understand the medium to long-term outcomes of different surgeries.
4. L198: What are the mid-term results of this type of comparison compared to existing short-term studies?
5. L226: In the discussion section, we can discuss in more detail the anatomical or biomechanical mechanisms underlying the instability of HAM control rotation.
6. L240: It is mentioned in the article that "the gradient can achieve better biomechanical function in a better heating environment", so it would be more perfect if imaging evaluation, even histological evaluation in animal experiments, could be included in subsequent studies.

Reviewer 4 ·

Basic reporting

This paper retrospectively analyzed the clinical outcomes of 165 patients after ACL-Reconstruction, compared the effects of different anatomic femoral tunnel positions especially on anteroposterior and rotational stability, with more than 5 years follow up.
The manuscript is written in professional, objective language. The study was well designed and appropriate methods were applied. The data set and the conclusion is helpful for clinical work. However,there are still some details below that need to be mildly modified.
Due to above,my consideration on publication of this manuscript is minor revision before acceptance, without re-review.

Experimental design

The study was well designed as a retrospective research, meanwhile, methods has been described with sufficient detail.

Validity of the findings

The data set and the conclusion is helpful for clinical work.

Additional comments

Questions:
1、 Line64-66: ‘Pearle et al. (10) also proposed the concept of the IDEAL position for the femoral tunnel, which is an ideal position that is isometric and involves low graft tension.’ This explanation of the IDEAL position is not accurate enough. It may be described as“which is an ideal position that is a balance between isometric, direct insertion, eccentric, anatomic, and low tension”
2、 Line66-68: ‘Therefore, since the establishment of the anatomic theory for ACLR, in clinical practice, the position of the femoral insertion has changed from the anatomic center position to become more eccentric (11).’ There seems to be a logical confusion about the sentences underlined.
3、 Line84: ‘abnormal lower limb alignment;’, It is recommended to provide a more detailed description of the inclusion and exclusion criteria, such as ‘abnormal lower limb alignment (varus or valgus more than 5 degrees)’.
4、 Line108: It should be ‘intercondylar fossa’, not ‘femoral condyle’.
5、 Line108-110: Since being discussed with transtibial technique, the technique used in this study to create the femoral tunnel should be described more detailed.
6、 Line133-134: ‘Rotational knee laxity was measured by the PST’, what state of the patient is when the PST was conducted, conscious or anesthetized? Is the conscious state different between preoperative and postoperative, especially in 5 years follow up.
7、 In Table 4, there were 3 cases in HC group whose postoperative PST result was grade II positive, while in Table 5, there became 2 cases. Is there a failure case with low preoperative PST who proceed to high postoperative PST?
8、 Line156-157: In the RESULT section, the description is unclear and should be described as: the postoperative positive rate of PST is the highest in the HC group, followed by the LC group, and the lowest in the HAM group. There was a significant difference between the HC group and the HAM group, while there was no statistical difference between the HC group and the LC group, as well as between the LC group and the HAM group.
9、 Line173-174: ‘Compared with the LC group, the HAM and HC groups showed less anteroposterior knee laxity’. This description does not accord with the data in table 2 and 3.
10、 line177,198,211:The transtibial reconstruction technique, which has been mentioned many times, is a surgical technique for creating a femoral tunnel rather than a definition of the location of the femoral tunnel. However, the concept of anatomic reconstruction is a description of locating the femoral tunnel site in the ACL anatomic footprint area. The three groups compared in the article, HC, LC, and HAM, are three different femoral tunnel sites under the anatomic theory. Therefore, in the discussion section, it is not appropriate to directly compare and discuss the femoral tunnel location with the transtibial technique. Instead, comparison with the ‘overtop position’ (the femoral tunnel site usually created using transtibial technique) should be discussed properly. Considering that the transtibial technique is not closely related to this article, it is recommended to delete the relevant content to make the focus more on the anatomic location of the femoral tunnel.
11、 This paper mainly compares the localization of three different ACL-R anatomical position: LC, HC, and HAM, and a large amount of recent literature has shown that the IDEAL point is a better location. It is recommended to discuss the similarities and differences between these three anatomical position and IDEAL point by citing relevant literatures.
12、 Regarding rotational stability, in addition to selecting a more optimal location for the femoral tunnel, have the authors used the LET procedure to enhance rotational stability in the cases mentioned in this study, or in other cases? If so, what are the indications?
13、 Have some patients undergone revision surgery due to poor postoperative outcomes or undergone second-look operation for any reasons?
14、 The logicality of the DISCUSSION section needs to be greatly improved.

---

## Round 0.2 · accepted · Accept

Dear Dr. Zhou,

Thanks for the effort in the revision. Hereby confirm I that the authors have addressed all of the reviewers' comments. I have assessed the revision by myself, and I am happy with the current version. Therefore I would like to state that this manuscript is ready for publication.

Best,
Jian Song